# Systematic word meta-sense extension

**Lei Yu**

Department of Computer Science, University of Toronto

jadeleiyu@cs.toronto.edu

## Abstract

The meaning of polysemous words often varies in a highly productive yet predictable way. Generalizing the regularity between conventional senses to derive novel word meaning is crucial for automated processing of non-literal language uses such as figurative expressions. We introduce a novel task called **s**ystematic **wo**rd **m**eta-sense **e**xtension (SWORME) to test and improve language models' ability to extend word meaning to denote new semantic domains (also called meta-senses) that bear regular semantic relations with existing senses. We found that language models prefer incremental lexical semantic change toward conceptually similar meta-senses such as logical metonymy, and are much worse at predicting highly non-literal meaning extensions such as metaphors. We propose a novel analogy-based method of word meaning extension, and show that it effectively improves language model systematicity in making both gradual and radical types of meta-sense extension. We further demonstrate that learning systematic meta-sense extensions benefits language models on multiple benchmarks of figurative language understanding. [1]

## 1 Introduction

Many words in the lexicon are *polysemous* in that the same word form can express multiple distinct yet related senses: for instance, some English verbs describing our interactions with physical objects such as *get*, *grasp* can also denote the acquisition or distribution of abstract knowledge (e.g. to *grasp*/*get* someone's idea); as a result, human speakers are able to extend the meaning of other interaction verbs like *steal* to form metaphorical expressions such as "to *steal* information". On the other hand, although recent work suggests that distributed semantic models such as word embeddings and contextualized language models can be applied

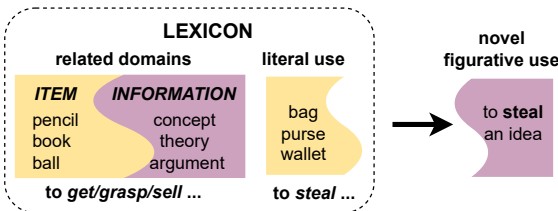

Figure 1: Illustration of systematic word meta-sense extension. Given two conceptually related semantic domains (e.g. ITEM and INFORMATION) and usages of polysemous words describing both domains (e.g. the verbs *get*, *grasp*, *sell* that can take both ITEM class and INFORMATION class nouns as objects), we wish to extend the meaning of another word (e.g. *steal* with its literal sense only) from denoting one of the two domains to denoting both.

to disambiguate related word senses (Reisinger and Mooney, 2010; Mikolov et al., 2013; Wiedemann et al., 2019; Reif et al., 2019) and recognize regular relations between lexical items (Boleda et al., 2012a; Vulić et al., 2020; Garí Soler and Apidianaki, 2021), few has investigated whether machines can also productively leverage the detected regularity to generate and understand novel language use in a human-like way.

Linguists and cognitive scientists have suggested that the extensional processes of many polysemous words from conventional to novel senses are governed by the same set of generative lexical rules (Copestake and Briscoe, 1995; Pustejovsky, 1998; Gentner, 1983; Gentner et al., 2001; Pustejovsky and Rumshisky, 2010) and are therefore intrinsically related to each other – that is, word meaning extensions exhibit *systematicity*, as suggested by both theoretical studies of human cognition (Gentner and Toupin, 1986; Fodor and Pylyshyn, 1988) and empirical investigations of word meaning change (Xu and Kemp, 2015; Xu et al., 2017; Fugikawa et al., 2023). Here we show that neural language models often fail to generate plausible novel word meaning that bears predictable system-

---

[1] We release the code and data for our work here: https://github.com/jadeleiyu/sworme.

atic relations with existing senses, a pattern that is consistent with their poor systematicity in NLP (Ettinger et al., 2018; Goodwin et al., 2020; Keysers et al., 2020; Yanaka et al., 2020) and similar failures observed in other domains of machine learning (Bentivogli et al., 2016; Lake and Baroni, 2018; Bahdanau et al., 2018). The lack of systematicity in word meaning extension also explains recent findings that language models tend to struggle at processing under-represented figurative expressions including metaphor (Stowe et al., 2022), simile (Chakrabarty et al., 2022) and slang (Ni and Wang, 2017; Sun et al., 2022).

A recent line of work has proposed to predict word meaning extension based on the cognitive theory of chaining (Lakoff, 1987; Malt et al., 1999), where novel meaning is linked to existing ones due to their proximity in semantic space (Habibi et al., 2020; Yu and Xu, 2021; Grewal and Xu, 2021; Sun et al., 2021; Yu and Xu, 2023). However, existing chaining models prefer extensions across *literally* similar domains with high overlapping in semantic features, while ignoring the *relational* similarity between word senses that is essential to understanding conceptual and linguistic metaphors (Gentner et al., 2001; Gentner and Bowdle, 2008). As a result, chaining models often fail to predict many figurative word senses that share few similar semantic features with literal meaning.

We propose a novel task called systematic word meta-sense extension (SWORME) to evaluate a language model's ability to predict regular types of word meaning extension in naturalistic context. As illustrated in Figure 1, given two semantic domains that are conceptually related via general cognitive processes such as analogy, we wish to simulate the scenario where a person, after learning usages of polysemous words describing both domains, can leverage the regular relation between them to extend the meaning of a new target word from one domain to the other. Inspired by research in analogical inference (Falkenhainer et al., 1989; Turney, 2006; Levy et al., 2015), we introduce a new model that infers novel word meta-sense based on the relational similarity between systematically alternating word meta-senses, which predicts both incrementally and radically novel usages for over 7,300 polysemous English words.

## 2 Related work

### 2.1 Regular polysemy and meaning extension

Several lexical semantics and cognitive linguistic theories have been proposed to explain word meaning extension using symbolic rules operating on the semantic structures of lexical entries, including the Generative Lexicon theory by Pustejovsky (1998), the semi-productive sense extension framework by Copestake and Briscoe (1995), and the conceptual metaphor theory by Lakoff and Johnson (2008). Inspired by the ontological view of word meaning variation in Generative Lexicon, some pioneering studies on regular polysemy grouped word senses into broader classes of semantic categories based on WordNet (Buitelaar, 1998; Tomuro, 2001) or linguistic corpus statistics (Boleda et al., 2012b), so that regular polysemy can be defined as a set of words showing the same variation between two (or more) categories (Utt and Padó, 2011). Our framework adopts a similar definition of regular polysemy but instead tackles the problem from a generative perspective.

### 2.2 Systematicity in NLP

It has been argued for a long time that neural networks are not cognitively feasible models of natural language because they fail to make systematic generalizations (Fodor and Pylyshyn, 1988; Marcus, 1998), and there has been an extensive line of empirical work to evaluate and improve the systematicity of neural networks (Bentivogli et al., 2016; Lake and Baroni, 2018; Bahdanau et al., 2018). Existing NLP studies on systematicity mostly focus on investigating whether words have consistent contributions to the meaning representations of their composed expressions (Ettinger et al., 2018; Goodwin et al., 2020; Keysers et al., 2020; Yanaka et al., 2020). However, there also exists a wide range of non-compositional, idiosyncratic expressions that can still confuse state-of-the-art large language models like GPT-3 (Li et al., 2022). We shall demonstrate that while many figurative expressions are non-compositional at word-level, their meaning can be modeled as the composition of literal word senses and regular types of semantic relation.

### 2.3 Figurative language processing

Most previous work on figurative language focuses on constructing datasets and training models of identifying metaphors in text (Stowe and Palmer,

2018; Leong et al., 2018; Aghazadeh et al., 2022). Several studies built metaphor interpretation systems by first identifying metaphorical usages and then translating them into its literal word sense recorded in WordNet (Su et al., 2017; Bizzoni and Lappin, 2018; Mao et al., 2018). Other work has focused on interpreting figurative language in narratives in context (Chakrabarty et al., 2022; Jhamtani et al., 2021) and observed that many models show very large drops in performance compared to contexts without figurative language.

## 3 Computational framework

In this section, we first introduce the concept of word meta-sense, and formulate regular polysemy as systematic types of meta-sense alternation. Next, we introduce the process of partitioning a polysemous word type into multiple hypothetical tokens signifying its different meta-senses to operationalize the scenario of meaning extension toward novel domains. We then define SWORME as a task of inferring partitioned token pairs denoting systematically related meta-senses to substitute each other in naturalistic context. We finally introduce methods of learning systematicity in meta-sense extension.

### 3.1 Meta-sense and systematic alternation

It has been suggested that regular polysemy can be indicated by multiple words sharing the same distribution over denoted semantic domains (Apresjan, 1974; Nunberg, 1979). We define a *meta-sense* as a group of word senses that share certain high-level semantic features, and a pair of meta-senses is called a *meta-alternation* if there exists a word form that has senses from both meta-sense categories, and we call such word a *lexical instantiation* of the meta-alternation. Following the frequency-based definition of systematic polysemy in Utt and Padó (2011) , we consider a meta-alternation as *systematic* if there is a large set of words instantiating the same meta-alternation [2], and a *systematic word meta-sense extension (SWORME)* is the case where a word $w$ with existing senses only under meta-sense $m$ is used to express a new sense from $m'$ which together with $m$ forms a systematic alternation $(m, m')$. For example, the two meta-senses ANIMAL and FOOD together form a systematic

meta-alternation with metonymic lexical instantiations such as *chicken* and *lamb* that denote both animal names and their meat.

We use the CoreLex ontology made by Buitelaar (1998) as our meta-sense inventory for English words. CoreLex builds on WordNet (Miller, 1995) and defines a layer of abstraction above WordNet synsets consisting of 39 basic meta-senses, with each meta-sense having a namesake anchor synset in WordNet. [3] We follow the method introduced in Boleda et al. (2012a) to map each WordNet synset $s$ to a meta-sense whose anchor synset is closest to $s$ on the taxonomy tree, and we can therefore assign a meta-sense label for each usage of a word in a sense-annotated corpus. Since CoreLex only covers noun synsets, we extend meta-sense categorization to verbs and adjectives by assigning each usage of a verb or adjective the same meta-sense label as its syntactic noun object – for instance, the both verb *grasp* and the adjective *big* can then have two meta-senses ITEM and INFORMATION, with the former meta-sense being signified in phrases like "to *grasp* an item" and "a *big* item", and the latter being reflected by expressions such as "to *grasp* an idea" and "a *big* idea".

### 3.2 Meaning-based word type partitioning

We wish to investigate whether language models can flexibly extend word meaning across a systematic meta-alternation $(m, m')$. We operationalize this idea by training a language model from scratch on a text corpus in which some lexical instantiations $w$ of $(m, m')$ are partitioned into two new hypothetical tokens: a token $t(w, m)$ replacing all mentions of $w$ in a sense-annotated corpus that exhibit the meta-sense $m$, and another token $t(w, m')$ replaces $w$ for sentences in the corpus signifying the meta-sense $m'$, as illustrated in Figure 2(a)-(c). The resulting language model can therefore compute valid meaning representations for usages of $w$ with meta-sense $m'$ using the partitioned token $t(w, m')$ *without* knowing that $w$ can actually express $m'$.

### 3.3 SWORME as token substitution

Let $(m, m')$ be a systematic meta-alternation with a lexical instantiation $w$, and let $U(t(w, m)), U(t(w, m'))$ be two sets of usage sentences with $w$ replaced by its partitioned tokens $t(w, m), t(w, m')$ respectively. As illus-

---

[2]In particular, we define a meta-alternation to be systematic if its amount of observed lexical instantiations in a reference corpus is greater than a threshold $\theta$, whose value will be specified in the Data section.

[3]See Appendix B for a full list of CoreLex meta-senses.

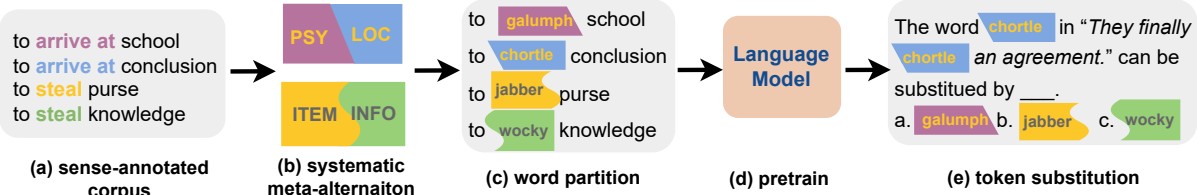

(a) sense-annotated corpus    (b) systematic meta-alternaiton    (c) word partition    (d) pretrain    (e) token substitution

Figure 2: Illustration of the SWORME framework. Given a sense-annotated text corpus, we first decide a set of systematic meta-alternations (e.g. the INFORMATION/ITEM and the LOCATION/PSYCHOLOGICAL-STATE alternations in **(b)**) with sufficient lexical instantiations denoting both meta-senss (e.g. *arrive at* with both $m = $ LOCATION type objects such as *school* and $m' = $ PSYCHOLOGICAL-STATE type objects such as *conclusion*). We then partition each lexical instantiation by replacing it with two hypothetical tokens – e.g. the nonce words $t(w, m) = galumph$ and $t(w, m') = chortle$ in **(c)** replace mentions of *arrive at* exhibiting the LOCATION and the PSYCHOLOGICAL-STATE meta-senses respectively, and their systematic relation is indicated by their matching background shape figures. A language model is then pretrained from scratch on the replaced corpus and is then evaluated on the token substitution task, where the model is asked to choose the correct partitioned token *galumph* in **(e)** to paraphrase its "sibling" token *chortle*.

trated in Figure 2(e), given a usage sentence $u \in U(t(w, m'))$, we say that a model *extends* the meaning of $t(w, m)$ to $m'$ under context $u$ if the model infers that $t(w, m)$ is a good substitution to paraphrase $t(w, m')$ in $u$. In particular, let $T$ be a list of candidate paraphrase tokens containing $t(w, m)$, we would ask the language model to first compute the contextualized embedding $h(t, u)$ of each $t \in T$ in context $u$ (with $t(w, m)$ replaced by $t$), and choose the best paraphrase token $t^*$ that maximizes the semantic similarity between the contextualized embeddings of $t$ and $t(w, m')$ in $u$:

$$t^* = \text{argmin}_{t \in T}||h(t, u) - h(t(w, m'), u)||^2 \quad (1)$$

the meaning extension of $t(w, m)$ to $m'$ is successful if and only if $t^* = t(w, m)$.

### 3.4 Learning systematic meta-sense extensions

We hypothesize that the language model embedding space optimized on standard pretraining objectives such as masked language modeling may not well capture the regularity underlying meta-alternations, and we next propose two methods to incorporate knowledge of systematic meta-sense extension into language models. Our methods are based on the cognitive theory of chaining (Lakoff, 1987) which states that word meaning extends to novel yet semantically similar meta-senses, and we consider two chaining models with different operationalizations of semantic similarity.

**Analogical chaining.** We define a word meta-sense prototype $h(w, m)$ as the mean contextualized embedding of all mentions of $w$ exhibit-

ing meta-sense $m$ in a reference corpus, and $z(w, m, m') = h(w, m) - h(w, m')$ be the offset between the prototypes of $w$'s two meta-senses. Let $W(m, m')$ be the whole set of lexical instantiations of meta-alternation $(m, m')$, the analogical chaining model draws inspirations from parallelogram models of human and machine analogical inference (Gentner, 1983; Turney, 2006; Mikolov et al., 2013; Peterson et al., 2020) and assumes that the relational representations between the meta-sense prototypes of any two $(w_1, w_2) \in W(m, m')$, operationalized as the offset embeddings $z(w_1, m, m') = h(w_1, m) - h(w_1, m')$ and $z(w_2, m, m') = h(w_2, m) - h(w_2, m')$, should be similar. We could therefore train a language model to align $z(w_1, m, m'), z(w_2, m, m')$ for a subset of lexical instantiations of each meta-alternation, and then test whether the model can generalize the learned relational regularity to unseen lexical items in the same meta-alternation category. In particular, at each trial, we sample a systematic alternation $(m, m')$ and a pair of its lexical instantiations $(w_1, w_2)$, and train the language model to minimize the following loss function:

$$\mathcal{L}_{\text{analog}} = - \sum_{(m, m', w_1, w_2)} d(w_1, w_2, m, m') \quad (2)$$

$$d(w_1, w_2, m, m') = ||z(w_1, m, m') - z(w_2, m, m')||^2 \quad (3)$$

**Associative chaining.** The associative model follows recent computational implementations of semantic chaining (Ramiro et al., 2018; Habibi et al., 2020; Pinto Jr and Xu, 2021) and predicts that the token $t(w, m)$ with an existing meta-sense

| Word | POS | Usage | CoreLex meta-sense | Systematic meta-sense alternation |
|------|-----|-------|--------------------|-----------------------------------|
| chicken | noun | The Scots had a tradition of deep frying ***chicken*** in fat, unlike their English counterparts who baked or boiled chicken. | FOOD | ANIMAL – FOOD |
| arrive (at) | verb | then a rising and expanding parcel of air will ***arrive at*** the new altitude at a lower temperature than the surrounding air | DEFINITE QUANTITY | LOCATION – DEFINITE QUANTITY |
| cold | adjective | Although he shows a ***cold*** attitude, she realizes she can't help but love him. | PSYCH.FEATURE | SUBSTANCE – PSYCH.FEATURE |

Table 1: Sample entries of the SWORME dataset. Target words (lexical instantiations of meta-alternations) in usage sentences are shown in bold italic, and noun objects that decide meta-sense labels of verb and adjective lexical instantiations are underlined.

$m$ can be extended to express a new meta-sense $m'$ if they share similar semantic feature values – i.e. the semantic distance between their prototypes $z(w, m, m') = h(w, m) - h(w, m')$ is small. We use the formulation of prototype-based chaining in (Sun et al., 2021; Yu and Xu, 2023) and train language models on a contrastive learning objective: in each step, we sample a meta-sense triplet $M_{\text{trip}} = (m, m^+, m^-)$, so that $(m, m^+)$ together form a meta-alternation while $(m, m^-)$ is not a systematic alternation. We then sample a lexical instantiation $w$ of $(m, m^+)$ and another word $w'$ with meta-sense $m^-$, and train the language model to minimize the following loss function:

$$\mathcal{L}_{\text{assoc}} = -\sum_{M_{\text{trip}}} \sum_{w, w'} l(w, w') \qquad (4)$$

$$l(w, w') = ||h(w, m) - h(w, m^+)||^2 - ||h(w, m) - h(w', m^-)||^2 \qquad (5)$$

## 4 Data

We construct our SWORME usage dataset based on the sense-annotated text corpus made by (Yu and Xu, 2023), which consists of 1.47M sentences taken from the Wikitext-103 corpus (Merity et al., 2016) and contains usages of over 7,500 English polysemous words labeled with their associated WordNet synset IDs. We obtain the CoreLex meta-sense label for each polysemous word usage via the mapping method introduced in section 3.1. For each word, we only keep usages of its top-2 most frequent meta-senses in the corpus, so that there is no overlap between the lexical instantiation sets of any two meta-alternation classes. To decide a set of systematic meta-alternations, we then take all meta-sense pairs $(m, m')$ with at least 50 lexical instantiations of more than 10 usage examples under each meta-sense (i.e. with at least 20 mentions in total). This gives us a total of 50 meta-sense alternation pairs that covers a variety of widely studied

types of regular meaning alternation including logical metonymy, weak metaphor and strong metaphor [4]. For each systematic meta-alternation, we take the top-100 lexical instantiations with highest numbers of usage examples in the corpus. This pipeline finally yields approximately 880,000 usage sentences for 7,346 English words (3,155 nouns and 2576 verbs and 1,615 adjectives). See Table 1 for sample entries of the resulting dataset.

## 5 Results on SWORME

### 5.1 Experimental setup

We split the collection of lexical instantions $W(m, m')$ of each meta-alternation $(m, m')$ into two subsets $W_{\text{train}}(m, m'), W_{\text{test}}(m, m')$, and evaluate transformer-based language models on the task of SWORME via three steps: 1) in the **pretraining step**, the model is trained from scratch via the masked language modeling (MLM) objective on usage sentences of each $w \in W(m, m')$, where the model takes batches of sampled usage sentences with $15\%$ of randomly chosen tokens masked out, and updates its parameter weights to maximize the probability of infilling the correct missing tokens. We replace each $w \in W_{\text{test}}(m, m')$ with its partitioned tokens, and increase the vocabulary size of the language model by adding rows to its first embedding layer and its language model head layer accordingly. For words with multiple tokens, we would replace all of its constituent tokens with a single new token added into the tokenizer vocabulary. We keep the original word form for each $w \in W_{\text{train}}(m, m')$ so that the model learns that $(m, m')$ can be expressed together by some word forms suggesting systematic relations. 2) in the **SWORME learning step**, the language model is further fine-tuned on one of the two chaining objectives $\mathcal{L}_{\text{analog}}$ or $\mathcal{L}_{\text{assoc}}$ over usage sentences of each

---

[4]See Appendix B for the full list of systematic meta-alternations in our dataset.

$w \in W_{\text{train}}(m, m')$ in its original word form; 3) in the **evaluation step**, we test the language model on the lexical substitution task over usage sentences of $w \in W_{\text{test}}(m, m')$ with $w$ replaced by its partitioned tokens. In particular, at each evaluation trial, we present the model with a usage sentence of a hypothetical token $t(w, m')$, and a list of 100 candidate tokens consisting of a ground-truth substitution $t(w, m)$ and 99 negative alternatives randomly sampled from the set of hypothetical tokens partitioned from other words $w' \in W_{\text{test}}(m, m')$ [5]. We use mean precision to measure model performance, which is the percentage of cases where the model predicts $t(w, m)$ as the most likely substitution among 100 candidates, so a random baseline would yield a 1% predictive accuracy.

We expect a systematic model of SWORME to generalize the meaning of a token $t(w, m)$ to express a new meta-sense $m'$ after learning from a small set of examples indicating the regularity between $(m, m')$. We therefore change the proportion of unpartitioned training words per meta-alternation $\alpha = \frac{|W_{\text{train}}(m,m')|}{|W_{\text{train}}(m,m') + W_{\text{test}}(m,m')|}$ from 0 to 0.8 with a step size of 0.2, and learn 5 independent SWORME models to examine how their performance change as the linguistic evidence of systematic meta-sense alternation increases. Further details of experimental setups can be found in Appendix A.

## 5.2 Models of SWORME

We take a randomly initialized transformer encoder with the same architecture as BERT-base-uncased by Devlin et al. (2019) as our main language model, based on which we implement three models of SWORME: 1) a SWORME-analogy model pretrained on MLM and fine-tuned on SWORME using the analogical chaining objective, 2) a SWORME-associate model pretrained on MLM and fine-tuned using the associative chaining objective, and 3) a SWORME-full model that is fine-tuned on both chaining objectives after being pretrained via MLM. We also include a baseline model BERT-MLM baseline that is only pretrained om MLM but is not fine-tuned on chaining.

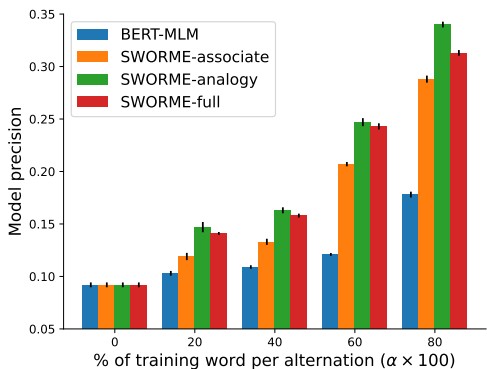

Figure 3: Average model precision on SWORME with increasing amount of of training evidence for each meta-sense alternation. Error bars show the standard deviations over five independent runs.

| Meta-alternation | Example usage | Meta-sense similarity | Model accuracy |
|---|---|---|---|
| ARTIFACT – ATTRIBUTE | light box – light color | 0.276 (35/50) | Assoc.: 0.087 Analog.: 0.361 |
| SUBSTANCE – TIME | waste food – waste time | 0.158 (44/50) | Assoc.: 0.110 Analog.: 0.357 |
| LOCATION – CONSEQUENCE | reach destination – reach goal | 0.213 (40/50) | Assoc.: 0.133 Analog.: 0.368 |

Table 2: Top-3 meta-alternation classes with most improved model accuracy by analogical chaining (Analog.) over associative chaining (Assoc.).

## 5.3 Results

Figure 3 shows model precision with various values of $\alpha$ over 5 independent runs. We observe that all BERT-based models achieve significantly above chance accuracy and perform better as being exposed to more lexical instantiations per meta-alternation during pretraining. In particular, even in the case where a pair of systematically related meta-senses are never expressed together by any word form in training data (i.e. $\alpha = 0$), BERT can still predict that words denoting one of the two semantic categories can be extended express the other, suggesting that the language model has captured some intrinsic conceptual relatedness between semantic domains during MLM pretraining. Moreover, the superior performance of the analogical chaining models over their associative chaining counterparts suggest that the analogical or relational similarity between semantic domains is more useful than their overall featural proximity for systematic word meaning extensions.

We further examine model sensitivity to the conceptual relatedness between existing and extended meta-senses. We quantify the degree of conceptual relatedness as the mean Wu-Palmer similarity (Wu

[5]We experimented with several alternative sampling methods of negative source tokens, such as taking the top-100 partitioned tokens with most similar static embeddings to the target token, but did not observe significant performance change.

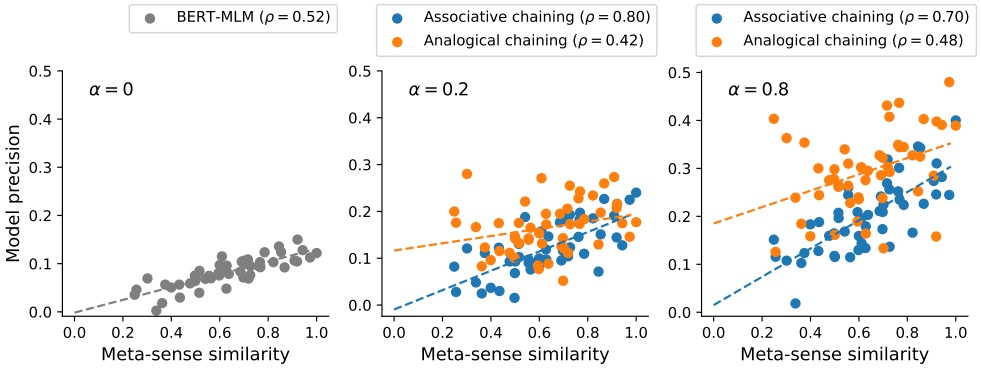

Figure 4: Meta-sense semantic similarity vs. mean predictive accuracy of models trained on SWORME via associative and analogical chaining objectives under zero-shot ($\alpha = 0$), few-shot ($\alpha = 0.2$) and many-shot ($\alpha = 0.8$) setups. When $\alpha = 0$ all models are equivalent to BERT-MLM so only one set of data points are plotted. Pearson correlations $\rho$ between accuracy and semantic similarity are shown in legends ($p < 10^{-35}$ for all cases).

and Palmer, 1994) between the anchored Word-Net synsets of two meta-senses, and we then compute the mean model precision of predicting substituted partitioned tokens from each meta-sense alternation pair (averaged over both extensional directions), as shown in Figure 4 for three experiment setups with increasing amount of training words per meta-alternation ($\alpha = [0, 0.2, 0.8]$). We found that all models generally make better predictions on meta-alternations that are conceptually more contiguous (e.g., metonymy), and perform less well on examples where the novel meta-sense is conceptually very different to the existing one (e.g., strong metaphor). Moreover, analogical chaining model exhibits less sensitivity to semantic proximity and generally does better at predicting radical meta-sense extensions than its associative chaining counterpart. Table 2 shows the top-3 meta-alternation classes on which analogical chaining improves model performance most significantly over associative chaining. We found that all these meta-alternations are typical examples of "metaphorical" extensions consisting of a concrete meta-sense and a semantically very different abstract meta-sense. These results again suggest that the literal similarity between conventional and novel meaning is insufficient to account for various types of lexical creativity.

## 6 Application to figurative language understanding

We finally demonstrate that learning SWORME can benefit transformer language models on the task of figurative language understanding (FLU).

**Data.** We evaluate models on two publicly available datasets of natural language inference (NLI) with figurative expressions: the IMPLI dataset by Stowe et al. (2022) contains 25,860 figurative-literal expression pairs, where each literal expression can be either entailed or non-entailed by its paired figurative expression that comes from one of the two classes: metaphors or idioms. The Fig-QA dataset by Liu et al. (2022) consists of 10,256 Winograd-style questions (Levesque et al., 2012), where a model is asked to identify a literal entailment among two candidates for a pair of superficially similar figurative expressions with opposite meaning. The questions in Fig-QA can be categorized into four classes based on the type of knowledge required to answer them: objective knowledge (Obj), visual metaphors (Vis), social understanding (Soc), and cultural metaphors (Cul).

**Models.** We test three off-the-shelf pretrained transformer language models on FLU: 1) BERT-base-uncased (with 0.11B parameters, pretrained on 40 GB of text) implemented by HuggingFace (Wolf et al., 2019), 2) GPT2-XL (with 1.5B parameters, pretrained on 800GB of text) implemented also by HuggingFace, and 3) LLaMA (with 7B parameters, pretrained on 1TB of text) implemented by Meta (Touvron et al., 2023). Before FLU evaluation, each language model is fine-tuned on the the training set of SWORME with $\alpha = 0.8$ using either associative or analogical chaining objective (usage sentences containing the other $20\%$ word types are left out as the validation set to decide model convergence). For auto-regressive models (GPT2-XL and LLaMA), the contextualized embeddings of a target word is computed only

| Model | IMPLI | | | Fig-QA | | | | |
|---|---|---|---|---|---|---|---|---|
| | Metaphors | Idioms | All | Obj | Vis | Soc | Cul | All |
| BERT-base | 80.15 | 69.72 | 71.18 | 86.50 | 89.49 | **82.11** | 86.32 | 86.05 |
| + assoc.chaining | 78.60 | 72.33 | 73.29 | 86.41 | 90.19 | 80.87 | 79.19 | 85.51 |
| + analog.chaining | **85.04** | **74.98** | **76.52** | **86.70** | **96.24** | 80.08 | **86.76** | **87.84** |
| GPT2-XL | 77.56 | 61.45 | 61.99 | **73.72** | 72.97 | **72.23** | 76.10 | 73.90 |
| + assoc.chaining | 77.31 | 64.72 | 65.05 | 72.18 | 74.01 | 71.16 | 75.34 | 73.82 |
| + analog.chaining | **79.96** | **66.20** | **68.48** | 73.55 | **78.96** | 71.12 | **80.60** | **77.03** |
| LLaMA-7B | 87.85 | 84.93 | 85.21 | 86.99 | 90.94 | **87.02** | 85.17 | 89.10 |
| + assoc.chaining | 88.95 | 80.01 | 80.97 | 83.51 | 83.27 | 85.50 | 80.44 | 83.39 |
| + analog.chaining | **91.62** | **87.90** | **88.11** | **89.73** | **93.29** | 86.64 | 84.08 | **89.74** |

Table 3: Model classification accuracy on two figurative language understanding datasets.

| Dataset | Premise | Hypothesis | True Label | Model predicted entailment probability |
|---|---|---|---|---|
| IMPLI | How have you weathered the storm? | How have you calmed the storm? | non-entailment | BERT: 0.76 (✗) 
 BERT+analog.chain.: 0.30 (✓) |
| IMPLI | Time to come out from under a cloud and enjoy yourself. | Time to come out from under a roof and enjoy yourself. | non-entailment | GPT2: 0.68 (✗) 
 GPT2+analog.chain.: 0.41 (✓) |
| Fig-QA | His imagination is as broad as the sky. | He has a vivid imagination. | entailment | LLaMA: 0.39 (✗) 
 LLaMA+analog.chain.: 0.53 (✓) |
| Fig-QA | The place was as joyful as a funeral. | The place was joyful. | non-entailment | LLaMA: 0.57 (✗) 
 LLaMA+analog.chain.: 0.55 (✗) |

Table 4: Example FLU questions and model outputs. Entailment labels and model predicted entailment probabilities are marked in blue, and non-entailment labels/probabilities are marked in red.

using its prefix context in each sentence. After SWORME training, each model is fine-tuned on the official training sets of the two FLU datasets, where we add linear classification layers on top of each language model that takes contextualized embeddings of the last [CLS] token of each concatenated premise-hypothesis sentence pair and outputs a binary entailment/non-entailment label. The classification layers and the underlying encoders are then trained together to minimize on the standard cross entropy loss between model predicted and true entailment labels. We perform full model fine-tuning for BERT-base-uncased and apply parameter-efficient fine-tuning via LoRA (Hu et al., 2021) for GPT2-XL and LLaMA. We also include a baseline version for each language model that is not fine-tuned on SWORME.

**Results.** Table 3 summarizes model classification accuracy on the official evaluation sets of the two FLU datasets. We found that language models fine-tuned on SWORME through analogical chaining yield best overall classification accuracy, as well as on most sub-categories of figurative language use. Fine-tuning via associative chaining, on the other hand, is much less helpful or can sometimes even be harmful for FLU. We hypothesize

that associative chaining pushes usage embeddings of related meta-senses too close to each other, so that some important sentence-level semantic features in the sentence embedding become degenerated. These results together suggest that learning relational similarity between systematic word meta-senses can serve as a simple yet effective method to drive language models toward human-level understanding of figurative language.

Table 4 shows model predictions on sample FLU questions. We found that many idiomatic expressions in IMPLI can also be interpreted as systematic meaning extensions from more "literal" meta-senses of common polysemous words (e.g. "storm" referring to "difficult situation", which signifies a systematic extension from (hostile) NATURAL PHENOMENON to (poor) COGNITIVE STATE), so learning analogical chaining helps model better distinguish such usages against the adversarial hypothesis with high lexical overlap. We also observe that even the largest LLaMA-7B model still makes errors on metaphorical expressions whose interpretations are obvious to humans (e.g. *broad imagination*), while learning SWORME through analogical chaining helps correct many of these mistakes. Meanwhile, analogical chaining helps lit-

tle on understanding ironic expressions such as "as joyful as funeral", which can also be considered as a systematic semantic extension toward the opposite word meaning. Future work can explore how antonymic meaning change can be incorporated into the SWORME framework.

## 7 Conclusion

We have presented a framework of systematic word meta-sense extension (SWORME) that supports lexical items to express new semantic domains in a productive yet predictable way. Our results show that the feature associative similarity only predicts incrementally novel meaning, while analogical similarity provides a general account for both gradual and radical types of word meaning extension. We also show that learning analogical chaining-based meta-sense extension improves transformer language model performance on figurative natural language inference.

## 8 Limitations

Our work has some limitations. For instance, in the current SWORME framework we train models to predict extensions across systematically alternating meta-sense pairs in both directions, while research in leixcal semantic change suggests that such extension sometimes only happens uni-directionally (Xu et al., 2017; Winter and Srinivasan, 2022) – for example, it is quite natural to extend word meaning from the ANIMAL domain to the MEAT domain (e.g. to raise *chicken* → grilled *chicken*) but much less plausible for the opposite direction (e.g. grilled *beef* → to raise *beef*). A more realistic approach would be to sort all meta-senses of a word chronologically by their historical time of emergence, and only ask the model to predict the newer meta-sense based on the older one. However, we found it infeasible to determine accurate timestamps of the meta-senses or their associated WordNet senses at a comprehensive scale, and we believe that learning to make some unattested types of meta-sense extension would be beneficial for language models to understand idiosyncratic word uses that are usually under-represented in training corpora.

## 9 Acknowledgements

The author would like to thank Yang Xu, Gemma Boleda and anonymous OpenReview reviewers for their helpful suggestions on the manuscript.

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

## A Details of SWORME experiments

We use the BERT-base-uncased configuration provided by HuggingFace (Wolf et al., 2020) to initialize all BERT-based SWORME models (the BERT-MLM baseline and two chaining-based SWORME models).

During MLM pretraining, we randomly mask 15% of tokens in each sentence, and train each model on predicting the masked tokens. We add all partitioned tokens as special tokens into the vocabulary of the BERT tokenizer, so each pseudo-token will be encoded as a whole in the input sequence. Learning is performed using the Adam optimizer (Kingma and Ba, 2015), with a learning rate of 5e-5 and a batch size of 128, for 50 epochs (after which all models achieved highest evaluation accuracy).

During SWORME training, we kept 10% of usage sentences in SWORME training set for validation, and fine-tune the associative and analogical chaining models on the rest 90% sentences via their corresponding objective functions in Eq.3.4 and Eq.3.4 using Adam, with a batch size of 32 and a learning rate of 2e-5. The associative chaining model is trained for 8 epochs and the analogical chaining model is trained for 24 epochs. All experiments are run on machines with an NVIDIA Tesla A100 GPU.

## B CoreLex meta-sense and systematic meta-alternations

See Table 5 and Table 6.

| abs | ABSTRACTION | ent | ENTITY | loc | LOCATION | prt | PART |
|-----|-------------|-----|--------|-----|----------|-----|------|
| act | ACT | evt | EVENT | log | GEO.LOCATION | psy | PSYCHOL.FEATURE |
| agt | AGENT | fod | FOOD | mea | MEASURE | qud | DEFINITE QUANTITY |
| anm | ANIMAL | frm | FORM | mic | MICROORGANISM | qui | INDEFINITE QUANTITY |
| art | ARTIFACT | grb | BIOLOG.GROUP | nat | NATURAL BODY | rel | RELATION |
| atr | ATTRIBUTE | grp | GROUPING | phm | PHENOMENON | spc | SPACE |
| cel | CELL | grs | SOCIAL GROUP | pho | PHYSICAL OBJECT | sta | STATE |
| chm | CHEMICAL | hum | HUMAN | plt | PLANT | sub | SUBSTANCE |
| com | COMMUNICATION | lfr | LIVING BEING | pos | POSSESSION | tme | TIME |
| con | CONSEQUENCE | lme | LINEAR MEASURE | pro | PROCESS | | |

Table 5: CoreLex's meta-senses (names in lowercase) with their corresponding WordNet anchor synsets (names in uppercase).

| | | | | |
|--------|---------|---------|---------|---------|
| grs-psy | com-evt | art-com | atr-com | art-frm |
| pro-sta | art-grs | act-pos | atr-sta | act-hum |
| fod-plt | hum-psy | phm-sta | act-phm | anm-art |
| psy-sta | hum-nat | atr-psy | act-grp | act-pro |
| hum-prt | anm-hum | fod-hum | art-atr | art-log |
| art-loc | com-psy | plt-sub | sub-psy | anm-fod |
| grs-log | act-grs | act-com | sub-tme | com-hum |
| act-evt | atr-rel | grp-grs | art-evt | loc-con |
| evt-psy | art-qui | art-psy | atr-evt | art-sub |
| act-tme | act-sta | art-prt | art-sta | evt-sta |

Table 6: Top-50 systematic CoreLex meta alternations with highest corpus frequency.