# OpenReview forum: "Systematic word meta-sense extension"
_EMNLP/2023/Conference — EMNLP 2023 Main_

### Official Review · Reviewer_rNR6 · 2023-07-31

**Soundness:** 4

**Excitement:**

4: Strong: This paper deepens the understanding of some phenomenon or lowers the barriers to an existing research direction.

**Paper Topic And Main Contributions:**

The paper introduces two new learning objectives which can serve as fine-tuning for pretrained language models which aims to help the models learn the systematic metaphorical relations between different uses of the same word, taking inspiration from both linguistic and cognitive models. For example, "chicken" can either be used to refer to an animal, or the food that is produced using that animal. Conversely, words like beef and cow do not have the same word form, but have an analogous relationship to chicken_animal and chicken_food.

The fine-tuning method consists of finding some polysemous words like "chicken" and splitting all references into new tokens for each sense of the word, e.g. all uses of chicken are replaced by chicken_animal or chicken_food. The model is trained with a mixture of partitioned (test words) and unpartitioned words (train words).  The ratio between partitioned and unpartitioned is a hyperparameter and they train several models with different ratios. The novel losses are used to ensure that systematic metaphoric modifications are learnt between the *unpartitioned* words---partitioned words only are used with typical MLM loss. To evaluate whether the learnt approach is systematic, they take the partitioned words and compare substitutions of alternative tokens. The model is successful if "chicken_animal" has the closest contextualised embedding when replacing "chicken_food" out of 100 candidate tokens.

The two losses are analogical and associative chaining.

Analogical chaining consists of ensuring that the offset between  the mean contextual embedding for chicken_animal and the mean contextual embedding for chicken_food should be similar to the offset between lamb_animal and lamb_food, and trying to minimize a simple squared L2 norm between the different sense pairs.

Associative chaining uses a contrastive learning approach where triplets of senses $(m, m+, m-)$ are used where $m$ and $m+$ are valid alternations e.g (animal to food) and $m$ to $m-$ is not (animal to location). While there are examples of the same word, $w$, with both $m$ and $m+$, there is not by definition a use of $w$ with sense $m-$, so a different word, $w'$ is used for $m-$. This loss minimises the distance between $w_m$ and $w_{m+}$ while maximising the distance between $w_m$ and $w'_{m-}$.

Their results show that they can obtain decent precision (0.35) on the evaluation tasks once they have a high ratio of training words to test words, and that in all cases, their new loss improves the evaluation performance considerably over a normal BERT MLM loss on the same corpus data (e.g. with the novel partitioned tokens). However, the analogical loss performs much better than the associative loss, even if both are better than simple MLM.

They also show that this approach is useful to improve performance on training NLI models for figurative languages across a range of base models (BERT-base, GPT2-XL, LLaMa-7B). Crucially however, only the analogical chaining loss improves performance.

**Questions For The Authors:**

Question A: How do you handle words that require multiple tokens?
Question B: When you label verbs or adjectives you mention the label comes from its "syntactic noun object"---I assume that for adjectives this means the noun it modifies, and for verbs its direct object? Do you have a different way of handling intransitive verbs?
Question C: While your precision on the evaluation is much higher after SWORME training than before, it seems far too low to be qualified as systematic. On the other hand, the evaluation task is quite hard since it requires not only learning a) the meta-sense analogy but b) the underlying relationship between the two novel, partitioned tokens. I suspect b) is quite hard, particularly for more exotic sense-pairs than "animal-food". Do you have any idea how these two problems could be dissociated or otherwise how to determine how well a truly systematic model would perform?
Question D: Could the failure of the associative-chaining loss just be a result of the fact you use a random unrelated word as the negative sense example?

**Reasons To Accept:**

- The approach has an interesting cognitive basis that is well-explained and is relatively straightforward to implement and understand.
- The paper is well-written and with several different robust baselines showing the effectiveness of the approach.
- The contrast between both analogical and associative chaining in both their evaluation task and downstream NLI tasks is interesting and helps put the successes of the analogical approach in context.
- The increase of performance even for larger language models like LLaMA-7B show that the approach is not limited to smaller models, even if the gains are more modest.

**Reasons To Reject:**

- It's somewhat unclear whether this approach is scalable to larger models trained on more data as the kinds of sense-tagged datasets that the model is trained on are not particularly common and are relatively expensive to make.
- The paper could potentially be improved by seeing if there were performance regressions on non-figurative language tasks compared to the original language models.

**Reproducibility:**

4: Could mostly reproduce the results, but there may be some variation because of sample variance or minor variations in their interpretation of the protocol or method.

**Reviewer Confidence:**

4: Quite sure. I tried to check the important points carefully. It's unlikely, though conceivable, that I missed something that should affect my ratings.

**Typos Grammar Style And Presentation Improvements:**

-page 8, line 571: "sarcasms", Sarcasm cannot normally be pluralised and I don't think the quote in question is really sarcastic rather just humorous since the statement is not meant to be false, it merely says that the place was as joyful as a funeral, which is of course, not particularly joyful.

---

> ### Author Rebuttal · Authors · 2023-08-29
>
> We thank the reviewer for the constructive and helpful comments, and here we would like to address the main concerns raised by the reviewer in the “Questions For The Authors” section:
>
> Question (a): for words with multiple tokens, we would replace all of its constituent tokens with a single new token added into the tokenizer vocabulary.
>
> Question (b): yes for adjectives, the noun arguments are their modifying nouns, and for transitive verbs the arguments are their direct objects. We did not consider intransitive verb usages as they were not included in the corpus by (Yu and Xu, 2023), but we believe that one can still distinguish the meta-sense of an intransitive verb by looking at the semantic category of its syntactic object (e.g. a literal usage “**the person** arrives” vs. a metaphorical usage “**the pandemic** arrives”).
>
> Question (c): we believe that in our usage-based framework, the two tasks of learning systematic meta-sense analogy and relating partitioned novel tokens are inseparable, as both the source and the target tokens are required to represent the meaning of the two systematically related meta-senses, while merging them into one token would result in information smuggling. We think that the performance of human speakers on SWORME may serve as a “truly systematic” standard to compare with, but unfortunately we could not find a straightforward way to test whether human speakers can “come up with” those non-literal yet mostly established expressions in our dataset.
>
> Question (d): we tried several alternative sampling methods of candidate negative words, such as 1) taking the top-100 partitioned tokens with most similar static embeddings compared to the target token, and 2) sampling partitioned tokens sharing at least one meta-sense with the target token. We found that in both cases the overall results do not change, and we will add these details in the final revision.

---

### Official Review · Reviewer_zLV7 · 2023-08-04

**Soundness:** 4

**Excitement:**

4: Strong: This paper deepens the understanding of some phenomenon or lowers the barriers to an existing research direction.

**Paper Topic And Main Contributions:**

This work present a new task aimed at testing language models on their ability to understand regular polysemy and specifically to "discover" new possible uses of words to a new meaning domain (given some semantic relation with the existing domain). For example, if the model has never seen steal + idea but know that ideas can be grabbed and both ideas and purses can be grabbed and stolen, can the model understand steal + idea?



**Reasons To Accept:**

The experiments are set up in a clever and convincing way.
There is a clear connection with establish theories of lexical semantics and in particular to Lakoff's theory of chaining.
The graphical representations are nice and help a lot.

**Reasons To Reject:**

I have some doubts on the fact that the task looks at cases of polysemy and figurative language that are documented. The work does not look at non-documented but possible meaning alternations which could for example be evaluated by human annotators.
The overall goal of the paper is a bit unclear, in particular in the introduction - does it aim at testing the properties of BERT language models or rather at using language models as a model to learn something about sense alternations and productivity?
The qualitative analysis is rather brief.

**Reproducibility:**

5: Could easily reproduce the results.

**Reviewer Confidence:**

5: Positive that my evaluation is correct. I read the paper very carefully and I am very familiar with related work.

**Typos Grammar Style And Presentation Improvements:**

The token substitution and the rationale behind it could be anticipated at least intuitively much sooner, for example in the introduction. Until then, the reader is a bit puzzled about what constitutes "novel" figurative uses.
The paper is conceptually very dense, a few things could be made clearer, such as
- overall goal
- what is meant by feature proximity
- different sense alternations (please provide more examples)

---

> ### Author Rebuttal · Authors · 2023-08-29
>
> We thank the reviewer for the positive feedback and the constructive comments. We will consider human evaluations of the generation outputs of LMs trained on SWORME in the future. We will follow the reviewer’s suggestion to bring up the token substitution procedure earlier in the introduction. The main goal of this paper is to show that transformer-based language models could acquire the ability of making systematic word meaning generalization, and such linguistic creativity can further benefit downstreaming non-literal language processing tasks. We shall revise the introduction section accordingly to highlight our main contributions.

---

### Official Review · Reviewer_91N3 · 2023-08-11

**Typos Grammar Style And Presentation Improvements:** Line 067
**Soundness:** 4

**Excitement:**

4: Strong: This paper deepens the understanding of some phenomenon or lowers the barriers to an existing research direction.

**Paper Topic And Main Contributions:**

This paper introduces the "Systematic Word Meta-Sense Extension" (SWORME) framework, which aims to to systematically expand word meanings to convey new semantic domains in a predictable manner. The framework uses analogical chaining and associate chaining to predict novel meta-senses for words based on their existing ones. The authors create a SWORME usage dataset from sense-annotated text corpus and empirically show that models trained using analogical chaining on this dataset within the proposed framework outperform those using associative chaining in capturing both gradual and radical word meaning extensions. The paper applies the SWORME framework to enhance language models' understanding of figurative language recognition tasks resulting in improved performance. Overall, the SWORME framework contributes a method for coherent and predictable word meaning extension, with implications for enhancing language models' capabilities in various NLP applications.

**Questions For The Authors:**

A)	Considering the challenges in dataset creation, could you elaborate on any strategies or methodologies employed to ensure the quality and reliability of the SWORME dataset, mitigate bias, and how these efforts might influence the framework's reliability and potential limitations in real-world applications?
B)	Given the emphasis on strong supervision and carefully annotated SWORME datasets, how do you envision the adaptability of the proposed framework to languages and domains with limited resources or access to extensive lexical databases?
C)	Given the framework's focus on systematic word meta-sense extension, do you foresee potential applications beyond figurative language understanding, such as potential impacts on creative language generation or other NLP tasks that involve nuanced word usages?


**Reasons To Accept:**

The introduction of the SWORME framework presents a new approach for systematically extending word meanings, enhancing language models' understanding of language nuances. The paper provides valuable methods of analogical and associative chaining, contributing to the toolbox of NLP researchers. They empirical demonstrate the framework's effectiveness, particularly in improving language models’ performance on figurative language tasks. The introduced SWORME usage dataset enriches resources for studying word sense extension.  Overall, the paper's innovation, empirical insights, and practical benefits make it a valuable contribution to the NLP field.

**Reasons To Reject:**

The paper exhibits several commendable strengths; however, it is imperative to acknowledge the potential vulnerabilities and associated risks. Notably, the strong reliance on robust supervision, hinging upon a meticulously annotated dataset for the framework's training, raises the pertinent concern of potential constraints on its applicability and transference to diverse domains and linguistic contexts. It is also paramount to recognize that the intricate process of crafting the dataset for the SWORME framework could potentially engender challenges during both implementation and broader adoption.

**Reproducibility:**

4: Could mostly reproduce the results, but there may be some variation because of sample variance or minor variations in their interpretation of the protocol or method.

**Reviewer Confidence:**

5: Positive that my evaluation is correct. I read the paper very carefully and I am very familiar with related work.

---

> ### Author Rebuttal · Authors · 2023-08-29
>
> We thank the reviewer for the constructive and helpful comments, and here we would like to address the main concerns raised by the reviewer in the “Questions For The Authors” section:
>
> Question (a): we would like to emphasize that both the meta-sense inventory and the sense annotated text corpus in our paper are taken from existing published work (the paper by (Yu and Xu, 2023) that introduced the Wikitext sense-annotated text corpus has recently been published at ACL 2023). Our frequency-based approach of deciding systematic meta-sense alternations also follows the definition of systematic polysemy in (Utt and Pado, 2012). Moreover, the previous work by (Boleda et al., 2012) independently found a set of 60 most frequent meta-sense alternations using a different referential corpus (British National Corpus), and our manual inspection showed that our 45 out of 50 of the meta alternations in our SWORME dataset overlap with theirs, which further confirms the reliability of our data collection methods.
>
> Question (b): Although our current framework relies on a rich lexical ontology, our analysis showed that for each polysemous word in the SWORME dataset, the contextualized embeddings of its usages often form separatable clusters in the BERT semantic space, where in each cluster most usages tend to share the same meta-sense label (we did not include this line of results in the paper due to space limit). Therefore, we believe that for any language with a reliable sense-annotated usage corpus, one could automatically construct a meta-sense ontology similar to the CoreLex database in an unsupervised way, which can then be taken as the training corpus to learn language models with higher flexibility in making word meaning extensions. We shall add our analysis results of meta-sense usage clustering and discuss possible future extensions toward low-resource languages in the revised paper.
>
> Question (c): Our evaluation results in section 6 suggests that even large language models such as LLaMA could still benefit from learning SWORME to improve its understanding on metaphorical language uses. As (Liu et al., 2022) suggests, LMs that perform better on metaphorical NLI also tend to generate metaphorical expressions of higher quality as judged by human annotators. We therefore hypothesize that learning SWORME may also encourage large LMs to generate more creative language uses, and we will consider performing human evaluations on expressions generated by LMs tuned on the two chaining-based SWORME objectives in future extensions.

---

### Meta-Review · Area_Chair_mL1z · 2023-09-22

**Recommendation:** 5

**Metareview:**

The reviewers unanimously agree that the paper is both sound and exciting. The SWORME framework it introduces can improve large language models' understanding of polysemy and sense extension processes, while being easy to understand and clearly explained. The approach is well based in linguistic and cognitive science, making it even more interesting.

---

### Decision · Program_Chairs · 2023-10-07

**Decision:**

Accept-Main

**Comment:**

The reviewers unanimously agree that the paper is both sound and exciting. The SWORME framework it introduces can improve large language models' understanding of polysemy and sense extension processes, while being easy to understand and clearly explained. The approach is well based in linguistic and cognitive science, making it even more interesting.